# Identification of miRNAs Involved in Maize-Induced Systemic Resistance Primed by *Trichoderma harzianum* T28 against *Cochliobolus heterostrophus*

**DOI:** 10.3390/jof9020278

**Published:** 2023-02-20

**Authors:** Shaoqing Wang, Xinhua Wang, Jie Chen

**Affiliations:** 1School of Agriculture and Biology, Shanghai Jiao Tong University, 800 Dongchuan Road, Shanghai 200240, China; 2State Key Laboratory of Microbial Metabolism, Shanghai Jiao Tong University, 800 Dongchuan Road, Shanghai 200240, China; 3Ministry of Agriculture Key Laboratory of Urban Agriculture (South), Shanghai Jiao Tong University, 800 Dongchuan Road, Shanghai 200240, China

**Keywords:** *Trichoderma harzianum*, maize, primed defense response, miRNA, *C. heterostrophus*

## Abstract

microRNAs (miRNAs) are known to play important roles in the immune response to pathogen infection in different plants. Further, *Trichoderma* strains are able to activate plant defense responses against pathogen attacks. However, little is known about the involvement of miRNAs in the defense response primed by *Trichoderma* strains. To explore the miRNAs sensitive to priming by *Trichoderma*, we studied the small RNAs and transcriptome changes in maize leaves that were systemically induced by seed treatment with *Trichoderma harzianum* (strain T28) against *Cochliobolus heterostrophus* (*C. heterostrophus*) infection in leaves. Through analysis of the sequencing data, 38 differentially expressed miRNAs (DEMs) and 824 differentially expressed genes (DEGs) were identified. GO and KEGG analyses of DEGs demonstrated that genes involved in the plant hormone signal transduction pathway and oxidation-reduction process were significantly enriched. In addition, 15 miRNA–mRNA interaction pairs were identified through the combined analysis of DEMs and DEGs. These pairs were supposed to play roles in the maize resistance primed by *T. harzianum* T28 to *C. heterostrophus*, in which miR390, miR169j, miR408b, miR395a/p, and novel miRNA (miRn5231) were more involved in the induction of maize resistance. This study provided valuable information for understanding the regulatory role of miRNA in the *T. harzianum* primed defense response.

## 1. Introduction

Plants’ small RNAs are a category of 20–24 nucleotide (nt) non-coding RNAs. They function critically in plant development and defense response by regulating target genes either through transcriptional or translational repression [1,2]. Moreover, accumulating evidence suggests that small RNAs play vital roles in immune responses to fungal pathogen attacks in different plants. miR160a, miR398b, miR166k-166h, miR169, miR164a, and miR1432, for instance, have been well studied for their roles in regulating rice immunity against *Magnaporthe oryzae* [3,4,5,6,7,8]. miR396, miR400, miR825, and miR773 have been revealed to be involved in the regulation of *Arabidopsis thaliana* resistance to necrotrophic pathogens—*Botrytis cinerea* and *Plectosphaerrella cucumerina*—and hemibiotrophic pathogens—*Fusarium oxysporum* and *Colletototrichum higginianum* [9,10,11,12]. In maize, overexpressing miR408b exhibited reduced resistance to *Fusarium verticillioides* infection in a susceptible maize line [13]. miR393b downregulated its putative target, the TIR1-like (F-box) gene, in maize leaf sheaths in response to *Rhizoctonia solani* [14]. Plant nucleotide-binding site leucine-rich repeat proteins (NBS-LRR) play important roles in the recognition of pathogens and defense response activation [15]. The miR482/2118 family has been found to target a large number of NBS-LRR genes, triggering the production of phased secondary small RNAs (phasiRNAs), which can act in cis or in trans to reinforce the silencing effects of miR482/2118 on NBS-LRR genes. This would confer resistance to *Verticillium dahlia in* cotton [16], potatoes [17], and *Fusarium oxysporum* in tomatoes [18]. In barley, the miR9863 family members targeted a subset of NBS-LRR genes to confer race-specific disease resistance against the powdery mildew fungus [19]. Similarly, in apples, miR390a and two novel miRNAs, miRLn11 and miRln20, directly targeted NBS genes to regulate the resistance to apple leaf spot disease caused by *Alternaria alternata* and *Glomerella cingulata* [20,21,22].

*Trichoderma*, as a well-known biocontrol agent (BCA) of plant pathogens, has been demonstrated to have a significant effect against a range of pathogen infections through different biocontrol mechanisms. It has been commonly confirmed that *Trichoderma* agents can activate maize-induced systemic resistance against foliar diseases through seed priming and root colonization [23,24,25]. Some studies have shown that the primed defense triggered by *Trichoderma* is mediated by a network of complex plant signaling pathways involving stimulated gene expression related to the host reactive oxygen species (ROS), salicylic acid (SA), jasmonic acid (JA), and ethylene (Et). In addition to the central players in defense, other hormones for plant growth are also simultaneously altered, such as auxin, gibberellin, and abscisic acid [26,27,28,29,30,31].

Emerging studies have focused on the role of miRNAs in induced systemic resistance (ISR) primed by beneficial bacteria. The miR825/miR825 * pair, miR472, miR846, and four candidate miRNAs (zma-miR169a-5p, zma-miR169c-5p, zma-miR169i-5p, and zma-miR395b-5p) were reported to regulate the ISR activated by different Bacillus spp., another well-known BCA [12,32,33,34,35]. Southern corn leaf blight (SCLB), caused by *Cochliobolus heterostrophus* (*C. heterostrophus*), is a major foliar fungal disease of maize leaves. Previous studies have reported that *Trichoderma* spp. can promote the biological control of SCLB by inducing defense-related genes and enzymes [36,37,38]. However, to our knowledge, little is known about the involvement, regulation, and function of miRNAs in *Trichoderma*-primed maize defense against SCLB.

To identify the miRNAs involved in the maize ISR primed by *Trichoderma*, we analyzed the expressions of miRNAs and the transcriptome in maize leaves, grown from seeds coated with *Trichoderma harzianum* (*T. harzianum*) T28 to protect against the challenging inoculation of *C. heterostrophus*, through Illumina Hiseq deep sequencing. *T. harzianum* T28 has previously been shown to be a good ISR activator in maize [28]. The results of this study broaden our knowledge to understand the *Trichoderma*-primed maize resistance at the miRNA level.

## 2. Materials and Methods

### 2.1. Plant Materials and Microbial Treatment

*T. harzianum* T28 and *C. heterostrophus* were cultured in potato dextrose agar (PDA) medium at 28 °C for 7 days, respectively. The spores of *T. harzianum* T28 were collected with a 1% sodium CMCase solution according to Yu et al. [28]. The spores of *C. heterostrophus* were collected by washing the plates with sterile water for maize leaf inoculation.

Maize B73 seeds were immersed in a 1% sodium CMCase solution (as a control), or a 1% sodium CMCase solution with *T. harzianum* T28 spores (1 × 10^6^/mL) (B73 + T28), according to Yu et al. [28]. They were then germinated at 28 °C for 3 days and then planted into pots (3 seeds/pot) under field conditions. After maize grew to the 6–8 leaf stage (about two months), the leaves were inoculated with *C. heterostrophus* (1 × 10^6^/mL), and the leaf samples were collected at 0 h, 6 h, 12 h, 24 h, 36 h, and 48 h, respectively. Each sample included six maize plants, and three replicates for each sample. All the samples were immediately frozen in liquid nitrogen or stored at −80 °C. Leaf samples at 12 h that were post-inoculated (hpi) with *C. heterostrophus* were further used for high-throughput sequencing.

After being inoculated with *C. heterostrophus* (1 × 10^6^/mL) for 3 days, maize leaves were photographed and the lesions were measured by Image J software.

### 2.2. Construction and Analysis of Small RNA Libraries and Transcriptome Libraries

Total RNA was extracted from B73 and B73 + T28 leaves, respectively, using the TRIzol Reagent, and each sample comprised three repeats. Then, six libraries (B73_1, B73_2, B73_3 and B73 + T28_1, B73 + T28_2, B73 + T28_3) for small RNA sequencing were constructed as described previously [22]. Briefly, small RNAs (18–30 nt) were isolated from the total RNA through PAGE gel separation, ligated with 5′ and 3′ adaptors, and transcribed to complementary DNA (cDNA). The amplified cDNA was sequenced using the Illumina HiSeq deep sequencing platform at Shanghai Personalbio Technology Co., Ltd. (Shanghai, China).

The raw reads were cleaned by removing low-quality and contaminated reads, and the clean reads were directly used for the following bioinformatics analysis. The clean reads were mapped to the maize B73 RefGen_V4 genome (https://maizegdb.org/genome/assembly/Zm-B73-REFERENCE-GRAMENE-4.0, accessed on 16 September 2016), the Rfam13 database (https://rfam.org/, accessed on 15 September 2017), and the miRBase database release22 (http://www.mirbase.org/, accessed on 12 March 2018) for miRNA identification. In addition, mireap software (https://sourceforge.net/projects/mireap/, accessed on 27 March 2013) was used to predict novel miRNA and minimum free energy (mfe) (kcal/mol) <−50.

The miRNA expression levels were normalized by reads per million and then analyzed using the edgeR seq method. The miRNAs with an expression fold change of ≥1.5 and a *p*-value < 0.05 were selected as differentially expressed miRNAs.

For transcriptome analysis, the same samples as those for small RNA sequencing were used to isolate total RNA with the TRIzol Reagent. Oligo(dT)-attached magnetic beads were used to purify mRNAs. Then, mRNAs were transcribed to complementary DNA (cDNA). The amplified cDNA was sequenced on the Illumina Hiseq2500 platform at Shanghai Personalbio Technology Co., Ltd. (Shanghai, China). After removing low-quality and contaminated reads, the remaining clean reads were directly used for comparisons with the maize B73 RefGen_V4 genome (https://maizegdb.org/genome/assembly/Zm-B73-REFERENCE-GRAMENE-4.0, accessed on 16 September 2016) using HISAT2 (http://ccb.jhu.edu/software/hisat2/index.shtml, accessed on 9 August 2015). The gene expression of each sample was calculated using the FPKM (fragments per kilo bases per million fragments) method. The differential analysis of gene expression was performed using DESeq software under the following conditions: expression difference log_2_FoldChange > 1 and a *p*-value < 0.05.

### 2.3. qRT-PCR Validation

Total RNA was reverse transcribed to first-strand cDNA using HiScript III RT SuperMix (+gDNA wiper) (Vazyme, Nanjing, China). The cDNA was further used for qPCR validation of the mRNAs using ChamQ Universal SYBR qPCR Master Mix (Vazyme, Nanjing, China). For the qRT-PCR validation of miRNAs, the total RNA was reverse transcribed to first-strand cDNA using a miRcute Plus miRNA First-Strand cDNA Kit (Tiangen Biotech, Co., Ltd., Beijing, China). Then, the cDNA was used for qRT-PCR validation using a miRcute Plus miRNA qPCR Kit (SYBR Green) (Tiangen Biotech, Co., Ltd., Beijing, China). The qRT-PCR analysis was performed using the LightCycler96 real-time PCR system (Roach). Gene expression levels were normalized using U6 rRNA and 18s rRNA as internal controls for miRNA and mRNA, respectively. Data were generated with three biological replications. The relative expression levels of mRNAs or miRNAs were calculated using the 2^−ΔΔct^ method. Primers used for qRT-PCR are listed in Appendix A.

### 2.4. Statistical Analysis

Student’s *t*-tests were performed to determine statistical significance by comparing the mean values from three independent replications using SPSS software (SPSS Inc., Chicago, IL, USA). The differences were deemed significant at a *p* < 0.05.

## 3. Results

### 3.1. Maize Resistance to SCLB Systemically Induced by T. harzianum T28

Maize B73 seeds were coated with *T. harzianum* T28 spores (B73 + T28) or without T28 (B73) and grew to the 6–8 leaf stage (Figure 1A). Maize leaves from B73 + T28 and B73 were inoculated with *C. heterostrophus* for 3 days, and the lesion areas were photographed and analyzed. Results showed that the leaf spots on B73 + T28 were significantly smaller than those on B73 (Figure 1B). Furthermore, we examined the expression levels of the marker genes of SA- (*PR4*), JA- (*LOX5*), and Et- (*ACO*) dependent signaling pathways, respectively (Figure 1C). Compared with the expression levels of these marker genes in B73, the expression levels of *LOX5* and *ACO*, rather than *PR4*, were induced at 6 hpi with *C. heterostrophus*; however, levels decreased at 12 hpi and subsequently increased again at 24 hpi. Further, the expression level of *PR4* increased and peaked at 24 hpi. These results showed that seed treatment with T28 was able to induce systemic resistance to SCLB.

### 3.2. Small RNA Profile Analysis in Maize B73 and B73 + T28 after C. heterostrophus Infection

In order to identify the miRNAs involved in the ISR primed by *T. harzianum* T28, maize leaves from B73 + T28 and B73 were inoculated with *C. heterostrophus* for 12 h respectively, and small RNA libraries were constructed from these maize leaves for high-throughput sequencing. In total, 38,234,791, 27,776,943, and 42,496,190 clean reads from B73 three replications, and 30,607,157, 37,190,750, and 22,554,099 clean reads from B73 + T28 three replications were obtained, respectively (Appendix A). The clean reads were then aligned with the databases of *Zea mays* B73 genome, Rfam 13, and miRbase 22 to identify rRNA, snRNA (small nuclear RNAs), snoRNA (nucleolar RNAs), tRNA, and known miRNA. The unmapped reads were further predicted for novel miRNA using mireap software (Appendix A). A total of 156 known maize miRNAs, belonging to 30 miRNA families, were identified (Appendix A). The number of each miRNA family member varied from 1 to 15. The zma-miR171 family, with 15 members, showed the most abundant miRNA family, followed by the zma-miR166 family, with 14 members. Furthermore, a total of 2009 potential novel miRNAs were identified (Appendix A). The length distribution of the small RNAs from the six libraries was analyzed. It was shown that the 24 nucleotide (nt) small RNAs were the predominant group, followed by 21 nt and 22 nt small RNAs in maize (Appendix A). The first nucleotide bias of small RNAs was also analyzed. For the canonical length of miRNAs (20–24 nt), a strong bias for U or G of the first nucleotide was observed (Appendix A).

### 3.3. Differentially Expressed miRNAs Responsive to ISR Primed by T. harzianum

A total of 38 differentially expressed miRNAs (DEMs), including 36 known miRNAs and 2 novel miRNAs, were identified (Figure 2A and Appendix A). Among these DEMs, 17 were upregulated and 21 were downregulated. zma-miR399j-5p and zma-miR156b-3p were expressed only in B73 + T28, and zma-miR395o-3p, zma-miR395b-5p, and two novel miRNAs (zma-m5231-5p and zma-m16709-3p) were expressed only in B73 (Figure 2A and Appendix A). To gain a better understanding of the regulatory roles of the known and novel maize miRNAs, target genes were predicted by psRNATarget software using the maize B73 RefGen_V4 genome (https://maizegdb.org/genome/assembly/Zm-B73-REFERENCE-GRAMENE-4.0, accessed on 16 September 2016). Further gene ontology (GO) and Kyoto Encyclopedia of Genes and Genomes (KEGG) pathway analyses showed that these target genes were enriched into phenylpropanoid and lignin catabolic process, inorganic anion transport, protein phosphorylation, and the regulation of intracellular signal transduction in the biological process (BP) category (Figure 2B). In the molecular function (MF) category, the GO terms were enriched into oxidoreductase activity, protein kinase activity, and transferase activity (Figure 2B). In the cellular component (CC) category, the GO term, the CCAAT-binding factor complex, was obviously enriched (Figure 2B). In the KEGG pathway analysis, these target genes were significantly enriched in sphingolipid and glycosphingolipid metabolism, ABC transporters, transcription factors, and protein export (Figure 2B).

### 3.4. Global mRNA Expression Profiles in Maize B73 and B73 + T28 after C. heterostrophus Infection

In order to identify the global gene expression alteration activated by *T. harzianum* T28, next-generation sequencing technology analyzed the transcriptome of the maize leaves from B73 + T28 and B73 that were inoculated with *C. heterostrophus* for 12 h. Overall, 46,484,604, 46,236,394, and 41,129,726 clean reads from three B73 replications, and 48,227,506, 49,683,436, and 51,121,654 clean reads from three B73 + T28 replications were obtained, respectively (Appendix A). Using HISAT2 alignment, more than 96% of the reads could be successfully mapped to the B73 genome, which covers more than 93% of genes.

### 3.5. Identification of Differentially Expressed Genes (DEGs) in Maize B73 and B73 + T28 after C. heterostrophus Infection

The expression levels of genes were normalized by FPKM (fragments per kilobases per million fragments). Under the criterion of a *p*-value ≤ 0.05 and |log2FoldChange| ≥ 1.0, a total of 824 DEGs were found, including 184 upregulated and 640 downregulated genes (Figure 3A and Appendix A). In order to explore the functions of these DEGs in maize responsive to the ISR activated by *T. harzianum* T28, GO and KEGG pathway analyses were performed. The GO analysis showed that these DEGs were significantly enriched into the categories of oxidation-reduction process (BP), oxidoreductase activity, and cofactor binding (MF) (Figure 3B). Interestingly, the KEGG pathway analysis showed that a significant number of DEGs fell into the category of plant hormone signal transduction (Figure 3C). The biological process of the oxidation-reduction category included 47 DEGs, of which, 13 DEGs were upregulated and 34 DEGs were downregulated (Figure 3D and Appendix A). A total of 26 DEGs were involved in the plant hormone signal transduction, including 10 upregulated DEGs and 16 downregulated DEGs (Figure 3E and Appendix A). In this category, there were 11 genes involved in the Auxin signaling pathway, 4 genes in the Cytokinin (CK) signaling pathway, 3 genes in Benzoxazinone (BX) synthesis, 2 genes in Ethylene (Et), Gibberellin (GA) signaling transduction, and Brassinosteroid (BR), respectively, and 1 gene in salicylic acid (SA) and jasmonic acid (JA) biosynthesis, respectively (Appendix A).

### 3.6. Combined Analysis of DEMs and DEGs

To further explore the regulatory functions of miRNAs involved in ISR, we compared the predicted target genes of DEMs with DEGs and analyzed the relationship between miRNAs and their target genes. Finally, 15 miRNA-mRNA pairs with opposing expression patterns were identified (Table 1). These 15 miRNAs belong to 7 families, zma-miR390, miR169, miR408, miR395, miR399, miR1432, and novel a miRNA, miRn5231. Among these miRNAs, miR169j, miR408b, miR395p, miR399c, and miR1432 each targeted two different genes, while three miRNAs (miR399b, miR399d, and miR399h) targeted the same gene.

Furthermore, to confirm the expression levels of ISR-associated miRNAs, we used quantitative, real-time PCR (qRT-PCR) to analyze the expressions of miRNAs and their target genes. The results showed that the expression levels of miR390b-5p, miR169j-5p, miR408b-3p, miR395p-3p, miR395a-5p, and miRn5231-5p were downregulated, and the expression levels of miR399d-3p, miR399c-5p, and miR1432-5p were upregulated (Figure 4A), while the target genes of these miRNAs showed opposing expression patterns (Figure 4B). The qRT-PCR results showed a consistent trend with the sequence data.

## 4. Discussion

The application of beneficial microbes, such as *Bacillus*, *Pseudomonas*, and *Trichoderma*, triggers an enhanced resistance state in the host, also termed induced systemic resistance (ISR), against a broad range of pathogens. This ISR capability is a consequence of reprogramming genes in plant different pathways. Emerging studies have shown that miRNAs play an important role in beneficial organism-primed ISR processes. Here, to identify the miRNAs and gene reprogramming in ISR primed by *T. harzianum* T28, we simultaneously analyzed the miRNA and gene expression alteration in B73 and B73 + T28 after *C. heterostrophus* infection. A total of 38 DEMs and 824 DEGs were identified. Through the interaction analysis between miRNAs and their target genes, 15 candidate ISR-associated miRNA–mRNA pairs were identified.

miR390 is a highly conserved and ancient miRNA family in land plants. Recently, it was shown that miR390 targeted transcripts encoding leucine-rich repeat (LRR) family proteins and LRR receptor-like serine/threonine-protein kinase (RPK) instead of its well-known target gene, TAS3 [39], playing a negative regulatory role in the defense response of apples against the fungal pathogen, *Alternaria alternata* [22]. LRR proteins and RPKs play an important role in plant defenses [40]. In our study, the expression of zma-miR390b-5p was reduced in B73 + T28, which had enhanced resistance to SCLB compared to B73 without T28 treatment, while its predicted target—the leucine-rich repeat protein kinase family protein (Zm00001d002835)—increased, thus implying the negative role of miR390b in maize defenses triggered by *T. harzianum* T28. The miR169/nuclear transcription factor Y subunit A (*NF-YA*) module plays important role in the development and abiotic stress resistance in various plant species [41,42,43,44,45]. In addition to its function as a regulator in abiotic stresses, miR169 could also participate in plant immunity. Osa-miR169 acts as a negative regulator in rice immunity against the blast fungus, *Magnaporthe oryzae* by repressing the expression of *NF-YA* genes [5]. In maize responses to *Bacillus velezensis* inoculation, three miR169 family members (zma-miR169a, zma-miR169c, and zma-miR169i) were all repressed, suggesting their potential roles in ISR [35]. In our study, the expression level of zma-miR169j-5p decreased, while its target genes, NF-YA (Zm00001d018255 and Zm00001d027874) were both upregulated in B73 + T28 post-inoculated with *C. heterostrophus*, consistent with that in the study completed by Xie et al. [35], thus implying that zma-miR169j-5p is involved in maize ISR primed by *T. harzianum* T28. Transgenic maize plants overexpressing miR408b exhibited reduced resistance to *Fusarium verticillioides* infection in a susceptible maize line [13]. In our study, the expression of zma-miR408b-3p and zma-miR408b-5p were all downregulated in B73 + T28, which showed enhanced resistance to *C. heterostrophus*, suggesting a negative role in ISR primed by *T. harzianum* T28. The homeodomain-leucine zipper transcription factors play crucial roles in regulating abiotic stress responses. Overexpression of the homeobox-leucine zipper protein, ATHB-6, improves the drought tolerance of maize [46]. In our results, homeobox-transcription factor 42 was upregulated, which might be the downregulation of a novel miRNA, zma-miRn5231-5p, thus suggesting zma-miRn5231-5p/Homeobox-transcription factor 42 involvement in maize defenses against *C. heterostrophus*. In rice, the amount of miR1432 was unchanged at 12 h post-inoculation (hpi) of *M. oryzae*, but decreased at 24 hpi and then significantly increased at 48 hpi. Blocking miR1432 led to enhanced blast disease resistance and increased yield in rice [7]. On the contrary, the expression of osa-miR1432 was induced under bacterial pathogen, *Xanthomonas oryzae pv. oryzae* (Xoo) strain PXO86, attack. Overexpressed osa-miR1432 heightened rice disease resistance to Xoo [47]. In our study, miR1432 was increased at 12 hpi of *C. heterostrophus* in B73 + T28, suggesting the involvement of miR1432 in maize immunity induced by T28 against *C. heterostrophus*. However, to explore whether miR1432 plays a negative or positive role in ISR against *C. heterostrophus*, further knock-down or overexpression of transgenic maize plants should be constructed.

The miR399/*PHOSPHATE2* (*PHO2*) pathway has been well studied in the regulation of Pi homeostasis [48]. Recently, the miR399/*PHOSPHATE*2 (*PHO2*) pathway has also been proven to play a positive regulatory role in response to freezing stress [49]. In addition, the Pi homeostasis regulated by miR399 has recently been shown to be involved in rice immunity responses against *M. oryzae* infection [50]. Moreover, in miR399-OE transgene maize lines, the genes for benzoxazinoid biosynthesis, *BX1*–*BX9*, were significantly downregulated, and the content of both 2,4-dihydroxy-1,4-benzoxazin-3-one (DIBOA) and 2,4-dihydroxy-7-methoxy-1,4-benzoxazin-3-one (DIMBOA), which are the predominant benzoxazinoids in plants [51], were also significantly reduced in miR399-OE plants [52]. Benzoxazinoids are a class of indole-derived plant metabolites that function in defense against numerous pests and pathogens [53]. These results demonstrate that miR399 functions as a negative regulator of plant immunity. However, in our study, 6 miRNAs belonging to the miR399 family were significantly upregulated in B73 + T28, which showed higher resistance than B73 without T28 treatment, and the predicted target genes were not the ortholog of *PHO2*. In addition, the genes for benzoxazinoid biosynthesis, *BX3,* and *BX13*, were all downregulated, while *BX12* increased in B73 + T28. Mutations in BXs biosynthesis genes (*Bx1*, *Bx*2, *Bx*6, and *Bx13*) increased maize resistance against *Exserohilum turcicum* infection [54]. *BX12* encodes O-methyltransferase, which catalyzes the conversion of DIMBOA-Glc to HDMBOA-Glc. Infection with fungal pathogens induced the methylation of DIMBOA-Glc to form HDMBOA-Glc in maize [55,56]. Thus, all these results implied that miR399 participated in maize ISR induced by *T. harzianum* T28 through a divergent defense strategy—possibly through an indirect mechanism with BX involved. Further exploration of miR399 knock-down transgenic plants coated with T28 is needed to reveal the mechanism of miR399 in maize defense responses to *C. heterostrophus.*

In maize, the function of miR395 has not been deeply studied. It is supposed to involve in the response to salt stress, drought stress, and virus infection by targeting the gene, *ATP sulfurylase* [57,58]. However, in our study, lipoxygenase7 (Zm00001d025524) was predicted to be a target gene of zma-miR395a-5p. The opposite expression trend was confirmed using qRT-PCR. Lipoxygenase 7 is involved in JA biosynthesis. It is widely accepted that ISR is caused by beneficial microbes, and the transmission signaling was commonly regarded to be JA/Et-dependent [59], while systemic acquired resistance (SAR) is generally considered to be induced by pathogenic microbes, and SA is essential in SAR signal transduction. Nevertheless, multiple reports demonstrating the activation of both SA and JA/Et signaling pathways in ISR triggered by *Trichoderma* revealed the complexity and diversity of signal pathways involved in ISR [60,61,62]. In this study, by detecting the lesion size of maize leaf spots and the expression of SA, JA, and Et marker genes in leaves, we could confirm that *T. harzianum* T28 was able to induce ISR in maize against *C. heterostrophus*, and SA, JA, and Et were all involved in the ISR, similar to the study conducted by Yuan et al. [60]. Further analysis of DEGs in plant hormone signal transduction, including two genes in Et, one gene in SA, and one gene in JA, also confirmed the above viewpoint.

In addition to SA and JA/Et, *Trichoderma* ISR is believed to involve a wider variety of signaling routes interconnected in a complex network of cross-communicating hormone pathways [24,63,64]. Cytokinin (CK), well known for its functions in controlling plant growth and development, can also affect plant resistance to disease. During *Botrytis cinerea* (*B. cinerea*) infection in tomatoes, the expression of CK response regulators and CK oxidase (CKX) genes were all significantly increased, and the expression location of CK-responsive genes followed the spread of infection. These results determined that the CK pathway is activated by pathogen infection [65]. They also confirmed that CK-ISR in tomatoes is an SA- and Et-dependent mechanism. Moreover, during *B. cinerea* infection in Arabidopsis, Ethylene receptor 1 can positively regulate general cytokinin signaling through CK-responsive type-A response regulators 2, which upregulates CK oxidase [66]. In our analysis of transcriptome data, the expression of both CK response regulators and CKX was up-regulated in B73 + T28, as was cytokinin hydroxylase, a gene involved in CK biosynthesis. Interestingly, one gene, Ethylene receptor 1, was also upregulated in B73 + T28, suggesting that the interconnected CK and JA signaling pathway might be involved in the ISR triggered by *T. harzianum* T28.

During the life cycle of plants, the trade-off between growth and defense is a dynamic process that is indispensable for plants to optimize resource allocation in response to various developmental cues and environmental challenges [67]. Brassinosteroids (BRs), Gibberellin, and Auxin are well-known phytohormones for regulating plant growth and development. In the plant dilemma between “to grow” and “to defend” in response to various stimuli, phytohormones rapidly activate crosstalk to facilitate the growth–stress tolerance trade-off in plants. BR catabolism leads to decreased levels of bioactive BRs to limit growth [68]. Gibberellin promotes the interaction between its receptor, GIBBERELLIN INSENSITIVE DWARF1 (GID1), and DELLAs and triggers the degradation of DELLAs through the ubiquitin/26S proteasome-dependent proteolytic pathway, thus activating GA-responsive growth and development in plants [67]. In our results, the gene for BR synthesis was downregulated and the gene for BR synthesis was upregulated. Further, DELLA proteins, GAI-like and GID 1, all decreased in B73 + T28. Moreover, the gene expressions of auxin synthesis and transport were alerted. Two genes related to auxin synthesis and two genes related to auxin response decreased in B73 + T28 after *C. heterostrophus* infection. These results implied that downregulation of this phytohormones-responsive growth in B73 + T28 during *C. heterostrophus* infection balanced the growth and ISR primed by *T. harzianum* T28. The regulatory roles and crosstalk of these phytohormones in ISR primed by *Trichoderma* need further exploration.

In conclusion, we identified 39 differentially expressed miRNAs and 824 DEGs in maize ISR primed by *T. harzianum* T28 to *C. heterostrophus* infection. A total of 15 miRNA–mRNA pairs were identified through combined small RNA and gene expression analysis, and these pairs are supposed to play significant roles in the maize ISR primed by *T. harzianum* T28 to *C. heterostrophus* (Figure 5). This study provided valuable information for understanding the regulatory roles of miRNAs in ISR triggered by *T. harzianum* and could help to develop a novel strategy for crops to protect against fungal pathogen infection. Next, we will attempt to exploit the real contribution of those selected differential expressed miRNA to the *Trichoderma*-based ISR against foliar disease by knocking out each of the DEMs individually or together.

## Figures and Tables

**Figure 1 jof-09-00278-f001:**
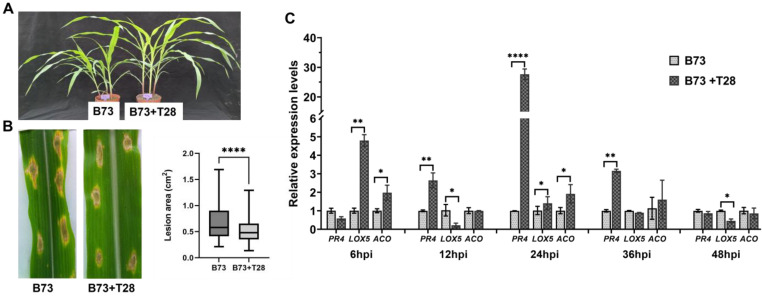
*Trichoderma harzianum* T28 primed induced systemic resistance (ISR) in maize against *Cochliobolus heterostrophus*. (**A**) The growth of maize with or without T28 coated seeds at the 6–8 leaf stage. Maize B73 seeds were coated with *T. harzianum* T28 (B73 + T28) or without T28 (B73, as a control) and grew to the 6–8 leaf stage at field conditions. (**B**) Disease severity against *C. heterostrophus* in maize B73 and B73 + T28. Maize leaves of B73 and B73 + T28, at the 6–8 leaf stage, were inoculated with *C. heterostrophus* for 3 days; the lesion spots were photographed, and the lesion areas of the spots were measured. Asterisks indicate statistical differences between treatments (Student’s *t*-tests; *p* < 0.05). (**C**) The expression levels of the defense-related genes (B73 + T28/B73) in maize leaves after inoculation with *C. heterostrophus*. PR4 (pathogenesis-related protein), LOX (lipoxygenases), and ACO (1-aminocyclopropane-1-carboxylate oxidase) are the marker genes of the salicylic acid-, jasmonic acid-, and ethylene-dependent signaling pathways, respectively. The expression levels of the defense-related genes were analyzed using qRT-PCR and normalized to 18s rRNA. Each sample contained 6 plants, and each treatment was repeated three times. Values represented means ± SD of three replicates.

**Figure 2 jof-09-00278-f002:**
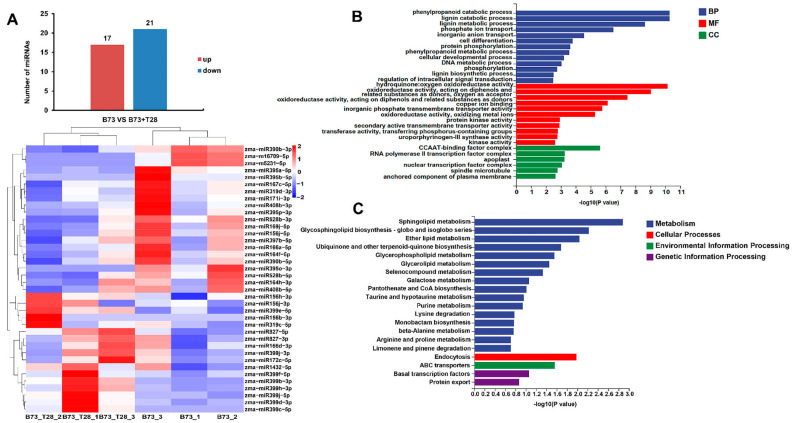
Differentially expressed maize miRNAs responsive to ISR primed by *T. harzianum* T28 (**A**) Statistics and heatmap of differentially expressed miRNAs (DEMs) in maize (B73 vs. B73 + T28). The expression levels of the differentially expressed miRNAs in B73 and B73 + T28 replicates were used to create the heatmap by employing the R package. The different color codes represented the different expression levels, ranging from −2 (blue) to 2 (red). Blue and red show low and high expression levels, respectively. (**B**) GO and (**C**) KEGG pathway analyses of the predicted target genes of DEMs.

**Figure 3 jof-09-00278-f003:**
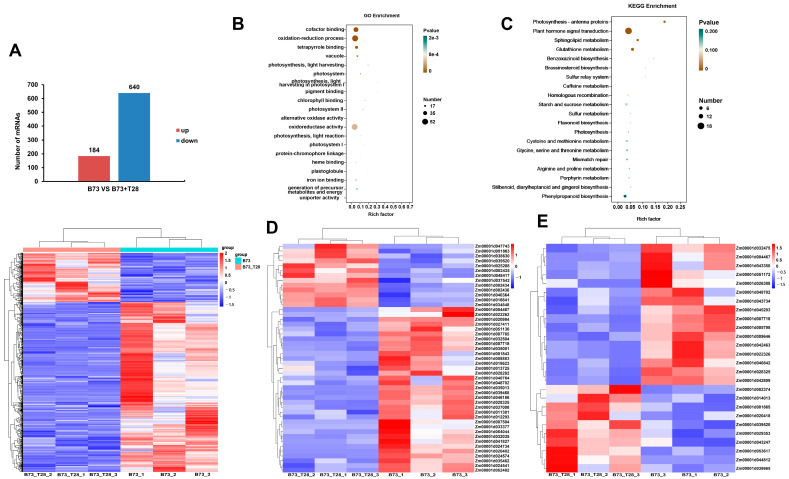
Differentially expressed genes in maize responsive to ISR primed by *T. harzianum* T28 (**A**) Statistics and heatmap of differentially expressed genes (DEGs) in maize (B73 vs. B73 + T28). (**B**) GO and (**C**) KEGG pathway analyses of the predicted target genes of DEMs. (**D**) Heatmap of DEGs in the biological process of the oxidation-reduction category and (**E**) the plant hormone signal transduction category. The expression levels of the differentially expressed genes in B73 and B73 + T28 replicates were used to create the heatmaps by employing the R package. The different color codes represent the different expression levels; blue and red show low and high expression levels, respectively.

**Figure 4 jof-09-00278-f004:**
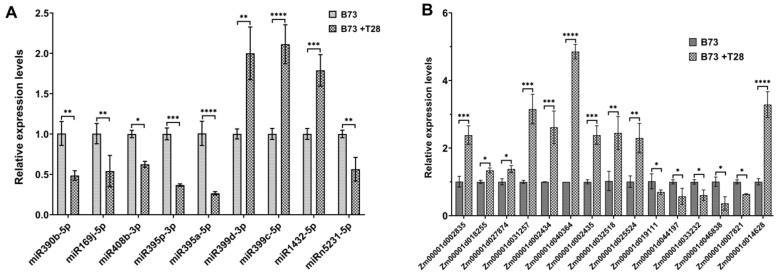
The expression levels of the miRNAs and target genes associated with ISR primed by *T. harzianum* T28. (**A**) qRT-PCR analysis of candidate ISR-associated miRNAs and (**B**) target genes of these candidate ISR-associated miRNAs in maize (B73 + T28/B73). Values represent the means ± SDs of the three replicates. *, **, *** and **** indicate significant differences.

**Figure 5 jof-09-00278-f005:**
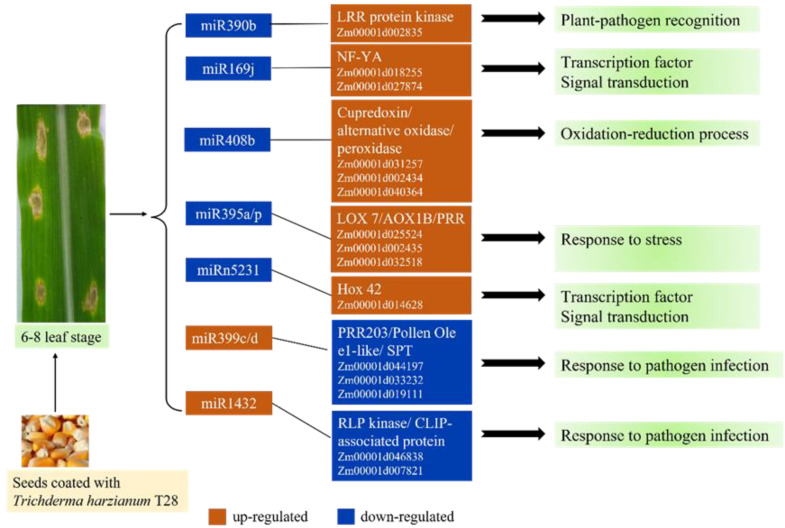
Potential regulatory roles of candidate ISR-associated miRNAs and their targets in maize primed by *T. harzianum* T28.

**Table 1 jof-09-00278-t001:** Potential miRNA/target pairs involved in maize ISR activated by *T. harzianum* T28.

miRNA	Foldchange (miRNA) (B73_T28/B73)	Target Gene	Foldchange (Gene) (B73_T28/B73)	Target_Start	Target_End	Score	Description
zma-miR390b-5p	0.63	Zm00001d002835	1.99	657	677	1	leucine-rich repeat transmembrane protein kinase family protein
zma-miR169j-5p	0.35	Zm00001d018255	1.66	1014	1034	2	nuclear transcription factor Y subunit A-3 isoform X5
zma-miR169j-5p	0.35	Zm00001d027874	2.18	1321	1341	2	nuclear transcription factor Y subunit A
zma-miR408b-3p	0.54	Zm00001d031257	1.55	850	870	2	Cupredoxin superfamily protein
zma-miR408b-3p	0.54	Zm00001d002434	2.62	1930	1950	3	alternative oxidase3
zma-miR408b-5p	0.66	Zm00001d040364	3.05	163	183	3.5	peroxidase
zma-miR395p-3p	0.59	Zm00001d002435	2.38	78	98	3	AOX1B, Ubiquinol oxidase
zma-miR395p-3p	0.59	Zm00001d032518	1.78	108	127	2.5	Pentatricopeptide repeat-containing protein
zma-miR395a-5p	0.06	Zm00001d025524	1.90	401	423	3.5	lipoxygenase7
zma-miR399d-3p, zma-miR399h-3p, zma-miR399b-3p, zma-miR399j-3p	4.01	Zm00001d019111	0.00	140	160	3.5	serine palmitoyltransferase
zma-miR399c-5p	Inf	Zm00001d044197	0.43	1843	1863	3	pentatricopeptide repeat protein 203
zma-miR399c-5p	Inf	Zm00001d033232	0.21	259	279	3.5	Pollen Ole e 1 allergen and extensin family protein
zma-miR1432-5p	1.58	Zm00001d046838	0.45	2562	2582	4	Putative receptor-like protein kinase family protein
zma-miR1432-5p	1.58	Zm00001d007821	0.00	706	726	4	Predicted CLIP-associated protein
zma-miRn5231-5p	0.01	Zm00001d014628	1.58	505	528	4.5	Homeobox-transcription factor 42

## Data Availability

The raw sequence data of small RNA and transcriptome reported in the current paper have been deposited in the Genome Sequence Archive at the National Genomics Data Center, Beijing Institute of Genomics, Chinese, Academy of Sciences/China National Center for Bioinformation (GSA: CRA009790) that are publicly accessible at https://ngdc.cncb.ac.cn/gsa, accessed on 29 January 2023).

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
