# Peer review of "Identification of miRNAs Involved in Maize-Induced Systemic Resistance Primed by Trichoderma harzianum T28 against Cochliobolus heterostrophus"

_jof, 2023, doi:10.3390/jof9020278_

Round 1

Reviewer 1 Report

The manuscript entitled "dentification of miRNAs involved in maize induced systemic resistance primed by Trichoderma harzianum T28 against Cochliobolus heterostrophus" aimed to study the small RNAs and transcriptome changes in maize leaves systemically induced by seed treatment with Trichoderma harzianum (strain T28) against C. heteostrophus infection in leaf. This study offered important insights into the regulatory role of miRNAs in the Induced Systemic Resistance (ISR) activated by T. harzianum. Moreover, this work expanded our knowledge to understand the Trichoderma-primed maize resistance at the miRNA level. The final product will be useful to researchers, and it could help to develop novel strategy for protecting crops from fungal pathogen infections.

Overall, the title and abstract are appropriate for the content of the text. The introduction is quite thorough and includes an adequate number of recent articles all the cited references relevant to the research. The experimental design is appropriate, and the results are clearly presented and discussed. This manuscript represents a generally well-written and well-organized article of miRNAs research.

I recommend that this paper be accepted in present form.

Best wishes

Author Response

Answer: Thank you very much for your comment.

Reviewer 2 Report

Reviewer’s comments

The MS describes the regulatory roles of miRNAs involved in induced maize resistance by T. harzianum. It gave strong evidences of differentially expressed miRNAs in primed maize which are able to limit pathogen effects caused by C. heterostrophus. I recommend to publish it

Material and Methods

Line 82: Please to precise how spores were collected, was the simple use of sterile water.

sufficient for isolation without presence of other propagules and medium (PDA) components.

Line 85 and 160: “coated with water” could better as ”water treated”.

Line 87: Please to give the age of the plants.

Line 130: Write Validation with small character.

Line 140: Please to give software used.

 Figures

Figure 1C: please to give statistical analyses (are there significant differences between values).

Figure 2: Please to write in clear characters, precise the software used for heatmap.

Figure 3: idem.

Figure 4: Please to give statistical comparison : significance of the differences between values.

Author Response

Material and Methods

Line 82: Please to precise how spores were collected, was the simple use of sterile water sufficient for isolation without presence of other propagules and medium (PDA) components.

Answer: The spores of T. harzianum T28 were collected with 1% sodium CMCase solution according to Yu et al. [28]. The spores of C. heterostrophus were collected through washing the plates with sterile water for maize leaf inoculation.

We have rephrased this in the new version of manuscript.

Line 85 and 160: “coated with water” could better as ”water treated”.

Answer: We have rephrased this in the new version of manuscript.

Line 87: Please to give the age of the plants.

Answer: Here, maize grew to the 6-8 leaf stage (about two months). We have rephrased this in the new version of manuscript.

Line 130: Write Validation with small character.

Answer: We have corrected this in the new version of manuscript.

Line 140: Please to give software used.

 Answer: We have reviewed this in the new version of manuscript.

 Figures

Figure 1C: please to give statistical analyses (are there significant differences between values).

Figure 2: Please to write in clear characters, precise the software used for heatmap.

Figure 3: idem.

Figure 4: Please to give statistical comparison : significance of the differences between values.

Answer: We have reviewed these figures in the new version of manuscript.